# DMSANet: Dual Multi Scale Attention Network

## Abstract

Attention mechanism of late has been quite popular in the computer vision community. A lot of work has been done to improve the performance of the network, although almost always it results in increased computational complexity. In this paper, we propose a new attention module that not only achieves the best performance but also has lesser parameters compared to most existing models. Our attention module can easily be integrated with other convolutional neural networks because of its lightweight nature. The proposed network named Dual Multi Scale Attention Network (DMSANet) is comprised of two parts: the first part is used to extract features at various scales and aggregate them, the second part uses spatial and channel attention modules in parallel to adaptively integrate local features with their global dependencies. We benchmark our network performance for Image Classification on ImageNet dataset, Object Detection and Instance Segmentation both on MS COCO dataset.

## 1 Introduction

The local receptive field of the human eye has led to the construction of convolutional neural networks which has powered much of the recent advances in computer vision. Multi scale architecture used in the famous InceptionNet (Szegedy et al., 2016) aggregates multi-scale information from different size convolutional kernels. Attention Networks has attracted a lot of attention recently as it allows the network to focus on only then essential aspects while ignoring the ones which are not useful (Li et al., 2019), (Cao et al., 2019) and (Li et al., 2019).

A lot of problems have been successfully tackled using attention mechanism in computer vision like image classification, image segmentation, object detection and image generation. Most of the attention mechanisms can be broadly classified into two types channel attention and spatial attention, both of which strengthens the original features by aggregating the same feature from all the positions with different aggregation strategies, transformations, and strengthening functions (Zhang et al., 2021).

Some of the work combined both these mechanism together and achieved better results (Cao et al., 2019) and (Woo et al., 2018). The computational burden was reduced by (Wang et al., 2020) using efficient channel attention and $1 \times 1$ convolution. The most popular attention mechanism is the Squeeze-and Excitation module (Hu et al., 2018b), which can significantly improve the performance with a considerably low cost. The "channel shuffle" operator is used (Zhang and Yang, 2021) to enable information communication between the two branches. It uses a grouping strategy, which divides the input feature map into groups along the channel dimension.

## 2 Related Work

There are two main problems which hinders the progress in this field: 1) Both spatial and channel attention as well as network using combination of two uses only local information while ignoring long range channel dependency, 2) The previous architectures fail to capture spatial information at different scales to be more robust and handle more complex problems. These two challenges were tackled by (Duta et al., 2020) and (Li et al., 2019) respectivly. The problem with these architectures is that the number of parameters increased considerably.

Pyramid Split Attention (PSA) (Zhang et al., 2021) has the ability to process the input tensor at multiple scales. A multi-scale pyramid convolution structure is used to integrate information at different scales on each channel-wise feature map. The channel-wise attention weight of the multi-scale feature maps are extracted hence long range channel dependency is done.

Non-Local block (Wang et al., 2018) is proposed to build a dense spatial feature map and capture the long-range dependency using non-local operations. (Li et al., 2019) used a dynamic selection attention mechanism that allows each neuron to adaptively adjust its receptive field size based on multiple scales of input feature map. (Fu et al., 2019) proposed a network to integrate local features with their global dependencies by summing these two attention modules from different branches.

Multi scale architectures have been used sucessfully for a lot of vision problems (?), (Hu et al., 2018b) and (Sagar and Soundrapandiyan, 2020). (Fu et al., 2019) adaptively integrated local features with their global dependencies by summing the two attention modules from different branches. (Hu et al., 2018a) used spatial extension using a depth-wise convolution to aggregate individual features. Our network borrows ideas from (Gao et al., 2018) which used a network to capture local cross-channel interactions.

The performance (in terms of accuracy) vs computational complexity (in terms of number of parameters) of the state of art attention modules is shown in Figure 1:

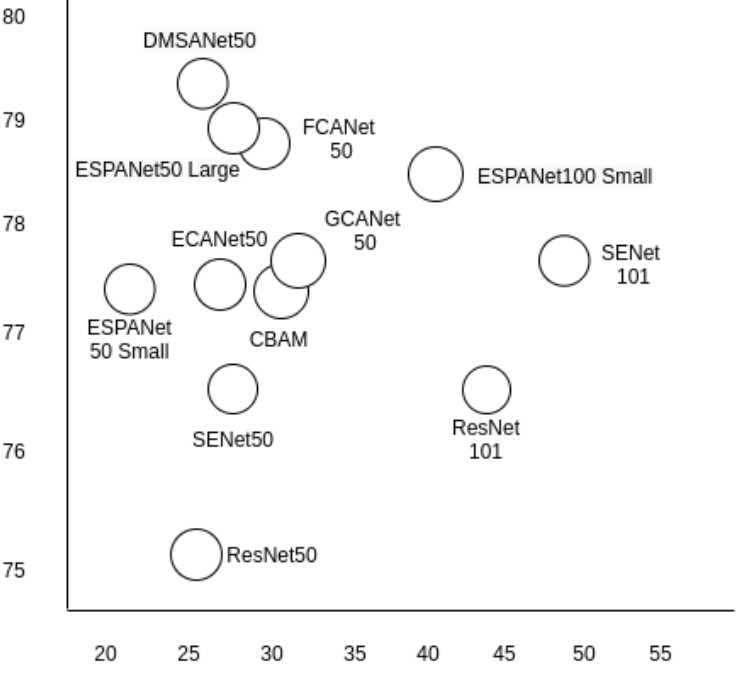

Figure 1: Comparing the accuracy of different attention methods with ResNet-50 and ResNet-101 as backbone in terms of accuracy and network parameters. The circles reflects the network parameters and FLOPs of different models. Our proposed network achieves higher accuracy while having less model complexity.

Our main contributions can be summarized as follows:

• A new attention module is proposed which aggregates feature information at various scales. Our network is scalable and can be easily plugged into various computer vision problems.

• Our network captures more contextual information using both spatial and channel attention at various scales.

• Our experiments demonstrate that our network outperforms previous state of the art with lesser computational cost.

## 3 METHOD

### 3.1 FEATURE GROUPING

Shuffle Attention module divides the input feature map into groups and uses Shuffle Unit to integrate the channel attention and spatial attention into one block for each group. The sub-features are aggregated and a "channel shuffle" operator is used for communicating the information between different sub-features.

For a given feature map $X \in R^{C \times H \times W}$ , where $C, H, W$ indicate the channel number, spatial height, and width, respectively, shuffle attention module divides $X$ into $G$ groups along the channel dimension, i.e., $X = [X1, X_G], X_k \in R^{C/G \times H \times W}$. An attention module is used to weight the importance of each feature. The input of $X_k$ is split into two networks along the channel dimension $X_{k1}, X_{k2} \in R^{C/2G \times H \times W}$. The first branch is used to produce a channel attention map by using the relationship of channels, while the second branch is used to generate a spatial attention map by using the spatial relationship of different features.

### 3.2 CHANNEL ATTENTION MODULE

The channel attention module is used to selectively weight the importance of each channel and thus produces best output features. This helps in reducing the number of parameters of the network. Let $X \in R^{C \times H \times W}$ denotes the input feature map, where the quantity $H, W, C$ represent its height, width and number of input channels respectively. A SE block consists of two parts: squeeze and excitation, which are respectively designed for encoding the global information and adaptively recalibrating the channel-wise relationship. The Global Average Pooling (GAP) operation can be calculated by the as shown in Equation 1:

$$GAP_c = \frac{1}{H \times W} \sum_{i=1}^{H} \sum_{j=1}^{W} x_c(i, j) \tag{1}$$

The attention weight of the $c^{th}$ channel in the SE block can be written as denoted in Equation 2:

$$w_c = \sigma \left( W_1 ReLU \left( W_0 \left( GAP_c \right) \right) \right) \tag{2}$$

where $W_0 \in R^{C \times C}r$ and $W_1 \in R^{Cr \times C}$ represent the fully-connected (FC) layers. The symbol $\sigma$ represents the excitation function where Sigmoid function is usually used.

We calculate the channel attention map $X \in R^{C \times C}$ from the original features $A \in R^{C \times H \times W}$. We reshape A to $R^{C \times N}$ , and then perform a matrix multiplication between A and the transpose of A. We then apply a softmax layer to obtain the channel attention map $X \in R^{C \times C}$ as shown in Equation 3:

$$x_{ji} = \frac{\exp \left( A_i \cdot A_j \right)}{\sum_{i=1}^{C} \exp \left( A_i \cdot A_j \right)} \tag{3}$$

where $x_{ji}$ measures the $i^{th}$ channel's impact on the $j^{th}$ channel. We perform a matrix multiplication between the transpose of $X$ and $A$ and reshape their result to $R^{C \times H \times W}$ . We also multiply the result by a scale parameter $\beta$ and perform an element-wise sum operation with $A$ to obtain the final output $E \in R^{C \times H \times W}$ as shown in Equation 4:

$$E_{1j} = \beta \sum_{i=1}^{C} \left( x_{ji} A_i \right) + A_j \tag{4}$$

### 3.3 SPATIAL ATTENTION MODULE

We use Instance Normalization (IN) over $X_{k2}$ to obtain spatial-wise statistics. A $Fc()$ operation is used to enhance the representation of $X_{k2}$. The final output of spatial attention is obtained by where

$W_2$ and $b_2$ are parameters with shape $R^{C/2G \times 1 \times 1}$. After that the two branches are concatenated to make the number of channels equal to the number of input.

A local feature denoted by $A \in R^{C \times H \times W}$ is fed into a convolution layer to generate two new feature maps $B$ and $C$, respectively where $B, C \in R^{C \times H \times W}$. We reshape them to $R^{C \times N}$, where $N = H \times W$ is the number of pixels. Next a matrix multiplication is done between the transpose of $C$ and $B$, and apply a softmax layer to calculate the spatial attention map $S \in R^{N \times N}$. This operation is shown in Equation 1:

$$s_{ji} = \frac{\exp\left(B_i \cdot C_j\right)}{\sum_{i=1}^{N} \exp\left(B_i \cdot C_j\right)} \tag{5}$$

where $s_{ji}$ measures the $i^{th}$ position's impact on $j^{th}$ position. Next we feed feature A into a convolution layer to generate a new feature map $D \in R^{C \times H \times W}$ and reshape it to $R^{C \times N}$. We perform a matrix multiplication between $D$ and the transpose of $S$ and reshape the result to $R^{C \times H \times W}$. We multiply it by a scale parameter $\alpha$ and perform a element-wise sum operation with the feature $A$ to obtain the final output $E \in R^{C \times H \times W}$ as shown in Equation 2:

$$E_{2j} = \alpha \sum_{i=1}^{N} \left(s_{ji} D_i\right) + A_j \tag{6}$$

## 3.4 Aggregation

In the final part of the network, all the sub-features are aggregated. We use a "channel shuffle" operator to enable cross-group information flow along the channel dimension. The final output of our module is the same size as that of input, making our attention module quite easy to integrate with other networks.

The whole multi-scale pre-processed feature map can be obtained by a concatenation way as defined in Equation 7:

$$F = \text{Concat}\left(\left[E_{1j}, E_{2j}\right]\right) \tag{7}$$

where $F \in R^{C \times H \times W}$ is the obtained multi-scale feature map. Our attention module is used across channels to adaptively select different spatial scales which is guided by the feature descriptor. This operation is defined in Equation 8:

$$att_i = \text{Softmax}\left(Z_i\right) = \frac{\exp\left(Z_i\right)}{\sum_{i=0}^{S-1} \exp\left(Z_i\right)} \tag{8}$$

Finally we multiply the re-calibrated weight of multi-scale channel attention $a_{tti}$ with the feature map of the corresponding scale $F_i$ as shown in Equation 9:

$$Y_i = F_i \odot att_i \quad i = 1, 2, 3, \cdots S - 1 \tag{9}$$

## 3.5 Network Architecture

We propose DMSA module with the goal to build more efficient and scalable architecture. The first part of our network borrows ideas from (Li et al., 2019) and (Zhang and Yang, 2021). An input feature map $X$ is splitted into $N$ parts along with the channel dimension. For each splitted parts, it has $C_0 = C_S$ number of common channels, and the $i^{th}$ feature map is $X_i \in R^{C_0 \times H \times W}$. The individual features are fused before being passed to two different branches.

These two branches are comprised of position attention module and channel attention module as proposed in (Fu et al., 2019) for semantic segmentation. The second part of our network does the following 1) Builds a spatial attention matrix which models the spatial relationship between any

two pixels of the features, 2) A matrix multiplication between the attention matrix and the original features. 3) An element-wise sum operation is done on the resulting matrix and original features.

The operators concat and sum are used to reshape the features. The features from the two parallel branches are aggregated to produce the final output. The complete network architecture is shown in Figure 2:

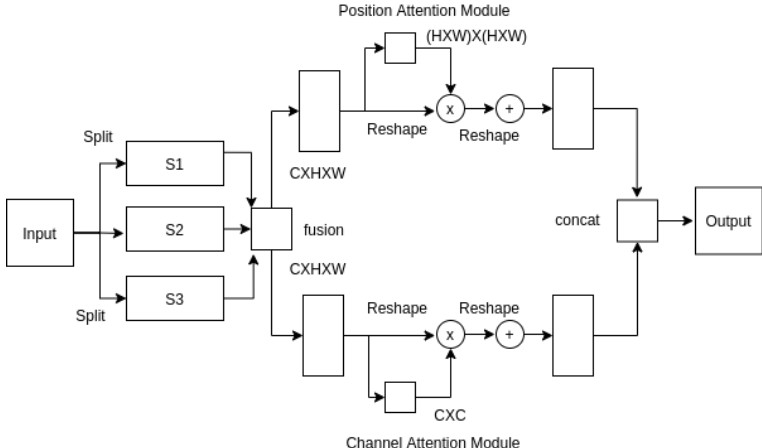

Figure 2: A detailed Illustration of DMSANet

.

We compare our network architecture with Resnet (Wang et al., 2017), SENet (Hu et al., 2018b) and EPSANet (Zhang et al., 2021) in Figure 3. We use our DMSA module in between $3 \times 3$ convolution and $1 \times 1$ convolution. Our network is able to extract features at various scales and aggregate those individual features before passing through the attention module.

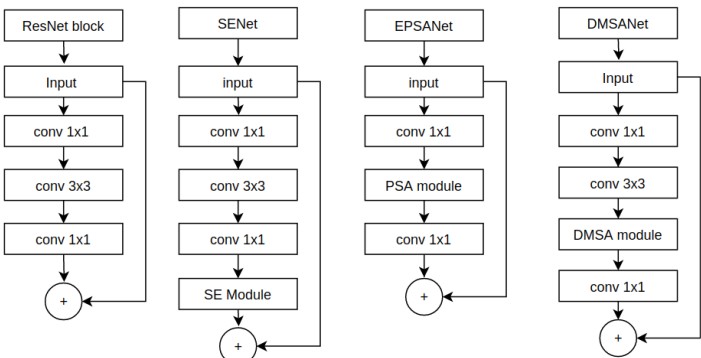

Figure 3: Illustration and comparison of ResNet, SENet, EPSANet and our proposed DMSANet blocks.

The architectural details our proposed attention network is shown in Table 1:

## 3.6 IMPLEMENTATION DETAILS

We use Residual Networks (He et al., 2016) as the backbone which is widely used in literature for image classification on Imagenet dataset (Deng et al., 2009). Data augmentation is used for increasing the size of the dataset and the input tensor is cropped to size $224 \times 224$. Stochastic Gradient Descent is used as the optimizer with learning rate of $1e^{-4}$, momentum as 0.9 and mini batch size of 64. The learning rate is initially set as 0.1 and is decreased by a factor of 10 after every 20 epochs for 50 epochs in total.

Table 1: Network design of the proposed DMSANet.

| Output | ResNet-50 | DMSANet |
|---|---|---|
| 112×112 | 7×7, 64 | 7×7, 64 |
| 56×56 | 3×3 max pool | 3×3 max pool |
| 56×56 | $\begin{bmatrix} 1\times1, & 64 \\ 3\times3, & 64 \\ 1\times1, & 256 \end{bmatrix} \times 3$ | $\begin{bmatrix} 1\times1, & 64 \\ DMSA, & 64 \\ 1\times1, & 256 \end{bmatrix} \times 3$ |
| 28×28 | $\begin{bmatrix} 1\times1, & 128 \\ 3\times3, & 128 \\ 1\times1, & 512 \end{bmatrix} \times 4$ | $\begin{bmatrix} 1\times1, & 128 \\ DMSA, & 128 \\ 1\times1, & 512 \end{bmatrix} \times 4$ |
| 14×14 | $\begin{bmatrix} 1\times1, & 256 \\ 3\times3, & 256 \\ 1\times1, & 1024 \end{bmatrix} \times 6$ | $\begin{bmatrix} 1\times1, & 256 \\ DMSA, & 256 \\ 1\times1, & 1024 \end{bmatrix} \times 6$ |
| 7×7 | $\begin{bmatrix} 1\times1, & 512 \\ 3\times3, & 512 \\ 1\times1, & 2048 \end{bmatrix} \times 3$ | $\begin{bmatrix} 1\times1, & 512 \\ DMSA, & 512 \\ 1\times1, & 2048 \end{bmatrix} \times 3$ |
| 1 × 1 | 7×7 GAP,1000-d fc | 7×7 GAP,1000-d fc |

We use Residual Network along with FPN as the backbone network (Lin et al., 2017a) for object detection. The detectors we benchmark against are Faster RCNN (Ren et al., 2015), Mask RCNN(He et al., 2017) and RetinaNet (Lin et al., 2017b) on MS-COCO dataset (Lin et al., 2014). Stochastic Gradient Descent is used as the optimizer with a weight decay of $1e^{-4}$, momentum is 0.9, and the batch size is 16 per GPU for 10 epochs. The learning rate is set as 0.01 and is decreased by the factor of 10 every 10th epoch.

For instance segmentation we use Mask RCNN (He et al., 2017) with FPN (Lin et al., 2017a) as backbone. Stochastic Gradient Descent is used as the optimizer with a weight decay of $1e^{-4}$, momentum is 0.9, and the batch size is 4 per GPU for 10 epochs. The learning rate is set as 0.01 and is decreased by the factor of 10 every 10th epoch.

## 4 RESULTS

### 4.1 IMAGE CLASSIFICATION ON IMAGENET

We compare our network with previous state of the art on ResNet with 50 and 101 layers.

Our network shows the best performance in accuracy, achieving a considerable improvement compared with all the previous attention models along with lower computational cost. The comparision of our network against previous state of the art with ResNet50 as backbone is shown in Table 2:

The comparision of our network against previous state of the art with ResNet101 as backbone is shown in Table 3:

### 4.2 OBJECT DETECTION ON MS COCO

The comparision of our network using Faster RCNN detector against previous state of the art is shown in Table 4:

The comparision of our network using MASK RCNN detector against previous state of the art is shown in Table 5:

The comparision of our network using RetinaNet detector against previous state of the art is shown in Table 6:

Table 2: Comparison of various attention methods on ImageNet with ResNet50 as backbone in terms of network parameters(in millions), floating point operations per second (FLOPs), Top-1 and Top-5 Validation Accuracy(%). The best records are marked in bold.

| Network | Params | FLOPs | Top-1 Acc (%) | Top-5 Acc(%) |
|---------|--------|-------|---------------|--------------|
| ResNet | 25.56 | 4.12G | 75.20 | 92.52 |
| SENet | 28.07 | 4.13G | 76.71 | 93.38 |
| CBAM | 28.07 | 4.14G | 77.34 | 93.69 |
| ABN | 43.59 | 7.18G | 76.90 | - |
| GCNet | 28.11 | 4.13G | 77.70 | 93.66 |
| AANet | 25.80 | 4.15G | 77.70 | 93.80 |
| ECANet | 25.56 | 4.13G | 77.48 | 93.68 |
| FcaNet | 28.07 | 4.13G | 78.52 | 94.14 |
| EPSANet(Small) | **22.56** | 3.62G | 77.49 | 93.54 |
| EPSANet(Large) | 27.90 | 4.72G | 78.64 | 94.18 |
| DMSANet | 26.25 | **3.44G** | **80.02** | **94.27** |

Table 3: Comparison of various attention methods on ImageNet with ResNet 101 as backbone in terms of network parameters(in millions), floating point operations per second (FLOPs), Top-1 and Top-5 Validation Accuracy(%). The best records are marked in bold.

| Network | Params | FLOPs | Top-1 Acc (%) | Top-5 Acc(%) |
|---------|--------|-------|---------------|--------------|
| ResNet | 44.55 | 7.85G | 76.83 | 93.48 |
| SENet | 49.29 | 7.86G | 77.62 | 93.93 |
| BAM | 44.91 | 7.93G | 77.56 | 93.71 |
| CBAM | 49.33 | 7.88G | 78.49 | 94.31 |
| SRM | 44.68 | 7.95G | 78.47 | 94.20 |
| ECANet | 44.55 | 7.86G | 78.65 | 94.34 |
| AANet | 45.40 | 8.05G | 78.70 | 94.40 |
| EPSANet(Small) | **38.90** | **6.82G** | 78.43 | 94.11 |
| EPSANet(Large) | 49.59 | 8.97G | 79.38 | 94.58 |
| DMSANet | 42.29 | 7.11G | **81.54** | **94.93** |

Table 4: Comparison of object detection results on COCO val2017 using Faster RCNN detector. The best records are marked in bold.

| Backbone | Params(M) | GFLOPs | AP | $AP_{50}$ | $AP_{75}$ | $AP_S$ | $AP_M$ | $AP_L$ |
|----------|-----------|--------|------|-----------|-----------|--------|--------|--------|
| ResNet-50 | 41.53 | 207.07 | 36.4 | 58.2 | 39.5 | 21.8 | 40.0 | 46.2 |
| SENet-50 | 44.02 | 207.18 | 37.7 | 60.1 | 40.9 | 22.9 | 41.9 | 48.2 |
| ECANet-50 | 41.53 | 207.18 | 38.0 | 60.6 | 40.9 | 23.4 | 42.1 | 48.0 |
| SANet-50 | 41.53 | 207.35 | 38.7 | 61.2 | 41.4 | 22.3 | 42.5 | 49.8 |
| FcaNet-50 | 44.02 | 215.63 | 39.0 | 61.1 | 42.3 | 23.7 | 42.8 | 49.6 |
| EPSANet-50(Small) | **38.56** | **197.07** | 39.2 | 60.3 | 42.3 | 22.8 | 42.4 | 51.1 |
| EPSANet-50(Large) | 43.85 | 219.64 | 40.9 | **62.1** | 44.6 | 23.6 | 44.5 | 54.0 |
| DMSANet | 44.17 | 222.31 | **41.4** | 61.9 | **46.2** | **25.8** | **44.7** | **55.3** |

## 4.3 INSTANCE SEGMENTATION ON MS COCO

We used Mask-RCNN (He et al., 2017) as benchmark on MS-COCO dataset (Lin et al., 2014). The comparision results of our network on instance segmentation using MS COCO dataset against previous state of the art is shown in Table 7:

Table 5: Comparison of object detection results on COCO val2017 using Mask RCNN detector. The best records are marked in bold.

| Backbone | Params(M) | GFLOPs | AP | $AP_{50}$ | $AP_{75}$ | $AP_S$ | $AP_M$ | $AP_L$ |
|---|---|---|---|---|---|---|---|---|
| ResNet-50 | 44.18 | 275.58 | 37.2 | 58.9 | 40.3 | 22.2 | 40.7 | 48.0 |
| SENet-50 | 46.67 | 275.69 | 38.7 | 60.9 | 42.1 | 23.4 | 42.7 | 50.0 |
| Non-local | 46.50 | 288.70 | 38.0 | 59.8 | 41.0 | - | - | - |
| GCNet-50 | 46.90 | 279.60 | 39.4 | 61.6 | 42.4 | - | - | - |
| ECANet-50 | 44.18 | 275.69 | 39.0 | 61.3 | 42.1 | 24.2 | 42.8 | 49.9 |
| SANet-50 | 44.18 | 275.86 | 39.4 | 61.5 | 42.6 | 23.4 | 42.8 | 51.1 |
| FcaNet-50 | 46.66 | 261.93 | 40.3 | 62.0 | 44.1 | 25.2 | 43.9 | 52.0 |
| EPSANet-50(Small) | **41.20** | **248.53** | 40.0 | 60.9 | 43.3 | 22.3 | 43.2 | 52.8 |
| EPSANet-50(Large)) | 46.50 | 271.10 | 41.4 | **62.3** | 45.3 | 23.6 | 45.1 | 54.6 |
| DMSANet | 47.23 | 279.26 | **43.1** | 61.6 | **47.5** | **24.1** | **46.9** | **56.5** |

Table 6: Comparison of object detection results on COCO val2017 using RetinaNet detector. The best records are marked in bold.

| Backbone | Params(M) | GFLOPs | AP | $AP_{50}$ | $AP_{75}$ | $AP_S$ | $AP_M$ | $AP_L$ |
|---|---|---|---|---|---|---|---|---|
| ResNet-50 | 37.74 | 239.32 | 35.6 | 55.5 | 38.2 | 20.0 | 39.6 | 46.8 |
| SENet-50 | 40.25 | 239.43 | 37.1 | 57.2 | 39.9 | 21.2 | 40.7 | 49.3 |
| SANet-50 | 37.74 | 239.60 | 37.5 | 58.5 | 39.7 | 21.3 | 41.2 | 45.9 |
| EPSANet-50(Small) | **34.78** | **229.32** | 38.2 | 58.1 | 40.6 | 21.5 | 41.5 | 51.2 |
| EPSANet-50(Large)) | 40.07 | 251.89 | 39.6 | 59.4 | 42.3 | 21.2 | 43.4 | 52.9 |
| DMSANet | 41.63 | 270.17 | **40.2** | **59.8** | **44.1** | **23.5** | **44.8** | **54.8** |

Table 7: Instance segmentation results of different attention networks by using the Mask R-CNN on COCO. The best records are marked in bold.

| Network | AP | $AP_{50}$ | $AP_{75}$ | $AP_S$ | $AP_M$ | $AP_L$ |
|---|---|---|---|---|---|---|
| ResNet-50 | 34.1 | 55.5 | 36.2 | 16.1 | 36.7 | 50.0 |
| SENet-50 | 35.4 | 57.4 | 37.8 | 17.1 | 38.6 | 51.8 |
| GCNet | 35.7 | 58.4 | 37.6 | - | - | - |
| ECANet | 35.6 | 58.1 | 37.7 | 17.6 | 39.0 | 51.8 |
| FcaNet | 36.2 | 58.6 | 38.1 | - | - | - |
| SANet | 36.1 | 58.7 | 38.2 | 19.4 | 39.4 | 49.0 |
| EPSANet-50(Small) | 35.9 | 57.7 | 38.1 | 18.5 | 38.8 | 49.2 |
| EPSANet-50(Large) | 37.1 | 59.0 | 39.5 | **19.6** | 40.4 | 50.4 |
| DMSANet | **37.4** | **61.1** | **40.7** | 19.3 | **40.9** | **51.7** |

## 4.4 ABLATION STUDY

The ablation studies of our architecture is shown in Table 8. The results are best obtained using instance normalization. Both removing $F_c()$ and using $1 \times 1$ Conv results in reduced performance as compared to the original network. The earlier is because $F_c()$ is used to enhance the performance of individual features while latter is because number of channels in each sub-feature is too few, so it is not important to exchange information among different channels.

## 5 CONCLUSIONS

In this paper, we propose a novel Attention module named Dual Multi Scale Attention Network (DMSANet). Our network is comprised of two parts 1) first for aggregating feature information at various scales 2) second made up of position and channel attention modules in parallel for capturing global contextual information. After evaluating our network both qualitatively and quantitatively, we show that our network outperforms previous state of the art across image classification, object

Table 8: Performance comparisons of our network using ResNet 50 as backbone with four options (i.e., original, using Batch Normalization, using Group Normalization, using shuffle normalization, eliminating $F_c()$ and using $1 \times 1$ Conv to replace $F_c()$ on ImageNet-1k in terms of GFLOPs and Top-1/Top-5 accuracy (in %). The best records are marked in bold.

| Methods | GFLOPs | Top-1 Acc(%) | Top-5 Acc(%) |
|---|---|---|---|
| origin | **3.44** | **80.02** | **94.27** |
| W BN | 3.82 | 77.37 | 93.80 |
| W GN | 3.56 | 77.61 | 92.89 |
| W SN | 3.51 | 78.16 | 93.48 |
| W/O $F_c()$ | 4.07 | 77.64 | 93.18 |
| $1 \times 1$ Conv | 3.55 | 78.69 | 93.71 |

detection and instance segmentation problems. The ablation experiments show that our attention module captures long-range contextual information effectively at various scales thus making it generalizable to other tasks. The best part of DMSANet attention module is that it is very lightweight and hence could be easily plugged into various custom networks as and when required.

ACKNOWLEDGMENTS

We would like to thank Nvidia for providing the GPUs for this work.

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
