# OpenReview forum: "DMSANET: DUAL MULTI SCALE ATTENTION NETWORK"
_ICLR.cc/2022/Conference — ICLR 2022 Submitted_

### Official Review · Reviewer_4y42 · 2021-10-25

**Correctness:** 3
**Technical Novelty And Significance:** 1
**Empirical Novelty And Significance:** 1
**Recommendation:** 3
**Confidence:** 5

**Main Review:**

Strengths:

Most of the proposed techniques are based on previous works, thus it is easy to follow. Three kinds of experimental verifications are provided, including image classification, object detection and instance segmentation. Better results are achieved on typical benchmarks.

Weaknesses:
1. Limited technical contributions
In fact, the proposed module largely borrows ideas from existing works, such as SENet, ResNet, shuffle attention, (Fu et al., 2019),  (Li et al., 2019) and (Zhang and Yang, 2021). I think this work is a combination of previous works. Most of the proposed modules have already been used in previous attention methods. The authors also admit these facts. Overall, I think this work is very limited in its novelty.

2. Insufficient evaluations
In this work, DMSA consists of many sub-modules. The authors list the ablations in Tab.8. However, there are no visual comparisons. Why the sub-modules are worked? How about their effects on typical image examples? In facts, the authors do not provide any visual examples to clarify the effects of the proposed attention module. Besides, there are no examples on image classification, object detection, instance segmentation. The performance improvements are very small when compared with other attention modules.

**Summary Of The Paper:**

In this paper, the authors propose a new attention module that shows better performance and lesser computation than most existing attention modules. Based on this module, the so-called Dual Multi Scale Attention Network is proposed. Several experiments are conducted to verify the performance on image classification, object detection and instance segmentation. Results shows that the proposed network has some advantages than previous works.

**Summary Of The Review:**

This work provides a new attention module to improve the performance of backbone networks. The main concerns are the limited techinical contributions and insufficient experiments. In my view, this work is much lower than the standard the ICLR.

---

### Official Review · Reviewer_gSEr · 2021-11-03

**Correctness:** 3
**Technical Novelty And Significance:** 2
**Empirical Novelty And Significance:** 2
**Recommendation:** 3
**Confidence:** 3

**Main Review:**

The strengths of the papers are as follow :

1-  Clarity of claims

2-  Extensive experiments

The weaknesses  are as follow :

1- Similarities and dissimilarities with most common related works are not enumerated. Namely, comparison with dual attention network for scene segmentation

2- One reference is not well formated. Absence of date and proceeding title.

3-  The paper claims that the proposed model captures well multi-scale information. However, no mention and comparison have been conducted w.r.t U-net models which are targeted to achieve multi-scale representation and trading-off local and global information.

4-  Figures and tables lack of information in the captions. Moreover, State-of-the-art methods are not referenced in the captions.

5-  Evaluation metrics for each task are not described.

6- Lack of analysis and discussion of the results. In the ablation study the table are not well discussed and analyzed. This the major drawback of the section


**Summary Of The Paper:**

The paper proposes a kind of multi-scale attention module capable at capturing spatial and channel attention informations considering long-range dependencies.

 The claims are corroborated with experimental studies for computer vision tasks, including Image classification and regression tasks.

**Summary Of The Review:**

Following the aforementioned consideration, l recommend to reject the paper "3: reject, not good enough" except if the authors make major changes, mainly providing an extensive analysis and discussion of the empirical results and considering U-net in the ablation study.

---

### Official Review · Reviewer_wSEj · 2021-11-07

**Correctness:** 3
**Technical Novelty And Significance:** 1
**Empirical Novelty And Significance:** Not applicable
**Recommendation:** 1
**Confidence:** 5

**Main Review:**

1. No novelty, not limited.  This is a simple combination of multi-scale and DANet.
2. Extremely poor paper organization, see Fig.1, Fig. 2, and Fig. 3.
3. Bad presentation, e.g, "Multi scale architectures have been used sucessfully for a lot of vision problems (?), (Hu et al., 2018b) and (Sagar and Soundrapandiyan, 2020)."  Even in this one sentence, there are a lot of mistakes. What is "(?)"? "sucessfully" -> "successfully"
4. Could authors explain why a simple "multi-scale + dual attention" idea could achieve such big improvements over all vision tasks? Based on my experience, that is impossible.



**Summary Of The Paper:**

In this paper, the authors proposed a new attention module, named Dual Multi-Scale Attention. Three commonly used methods are combined: multi-scale, channel-attention, and position attention. By introducing the new module (which is a simple combination without any novelty) to ResNet, it SIGNIFICANTLY improves the performance, e.g, 75.20 to 80.02  (and 76.83 to 81.54 for ResNet101) on ImageNet, 36.4 to 41.4 on MS COCO.

**Summary Of The Review:**

I would recommend strong reject for this paper, which introduces NO novelty and the experimental results are suspect.

---

### Official Review · Reviewer_QHkT · 2021-11-07

**Correctness:** 2
**Technical Novelty And Significance:** 1
**Empirical Novelty And Significance:** Not applicable
**Recommendation:** 1
**Confidence:** 4

**Main Review:**

**Methodology**
* The novelty is very limited. The proposed method is a simple combination of previous methods, and basically, no substantial new technique is introduced.

**(Unconvincing) Experiments**
* The experimental results are too good to be true. While the idea is simple and has been explored by many related works, authors still achieve unbelievable improvements over all these vision tasks.
* Also, the baseline is significantly lower than the original results: the image classification result of ResNet101 only achieves 76.83% in this paper, while the standard number should be around 77.37% (see torchvision models: https://pytorch.org/vision/stable/models.html)
* The results in Table 8 are confusing.  Why does removing FC result in larger GFLOPs?
* Could authors also report the training/inference efficiency? I assume it would be super slow.

**Presentation**
* The writing is rather poor, as well as the presentation. Too many typos. "combined both these mechanism", "respectivly", "The channel-wise attention weight of the multi-scale feature maps are extracted ...", "comparision", etc. These simple mistakes show this paper is far from well-prepared.

**Summary Of The Paper:**

This paper proposes a new method, Dual Multi Scale Attention (DMSA), as a plug-in module for CNN backbone. With the help of DMSA, authors claim to achieve huge improvements in image classification, object detection, and segmentation task.




**Summary Of The Review:**


Strong reject given the poor writing, limited novelty, and unconvincing experimental results of this paper.

---

### Decision · Program_Chairs · 2022-01-20

**Decision:**

Reject

**Comment:**

This paper is proposed to improve base CNN models by dual multi-scale attention module. To achieve a better feature representational ability, authors consider the multi-scale mechanism from both channel-dimension and spatial-dimension. The proposed method has been verified on several benchmarks, including ImageNet and MS COCO. However, all reviewers consider rejecting this paper because this work lacks novelty, the results are suspicious, and the writing is poor. No responses are submitted by authors to address the reviewers' concerns.